# Divergent representations of ethological visual inputs emerge from supervised, unsupervised, and reinforcement learning

## Abstract

Artificial neural systems trained using reinforcement, supervised, and unsupervised learning all acquire internal representations of high dimensional input. To what extent these representations depend on the different learning objectives is largely unknown. Here we compare the representations learned by eight different convolutional neural networks, each with identical ResNet architectures and trained on the same family of egocentric images, but embedded within different learning systems. Specifically, the representations are trained to guide action in a compound reinforcement learning task; to predict one or a combination of three task-related targets with supervision; or using one of three different unsupervised objectives. Using representational similarity analysis, we find that the network trained with reinforcement learning differs most from the other networks. Through further analysis using metrics inspired by the neuroscience literature, we find that the model trained with reinforcement learning has a high-dimensional representation wherein individual images are represented with very different patterns of neural activity. These representations seem to arise in order to guide long-term behavior and goal-seeking in the RL agent. Our results provide insights into how the properties of neural representations are influenced by objective functions and can inform transfer learning approaches.

## 1 Introduction

Many studies in machine learning aim to improve model quality or data efficiency by training the model with multiple objectives or transferring representations trained on other data. While leveraging other data or objectives may lead to higher data efficiency or increased generalization outside the primary task, relying too heavily on tasks other than the primary one can undercut performance if the other tasks are insufficiently well aligned. Humans and animals do in fact confront the full complexity of this continual learning problem of building representations, reusing what is worth transferring to new settings as they learn new things.

The potential utility of transferring representations is particularly salient in deep reinforcement learning, where rich sensory inputs must be transformed into actions. While landmark developments in deep RL have enabled the end-to-end training of agents capable of leveraging pixel-level visual information in complex environments to guide behavior (Mnih et al., 2015), there is a strong demand for methods that improve the sample efficiency of these training-intensive agents through pre- or joint-training of the representations they rely on. Indeed several studies aim to accelerate representation learning in vision-based RL through the use of additional objectives (e.g., Jaderberg et al., 2017; Wayne et al., 2018; Schwarzer et al., 2021).

In the present work, we examine a specific instance of a high-dimensional embodied control problem, the virtual rodent of Merel et al. (2020), which is also included in the RL Unplugged datasets for offline learning (Gulcehre et al., 2020). This problem involves visuomotor control of a high-dimensional body to solve multiple tasks, and it is solvable using a visuomotor policy trained through deep RL. In particular, we will compare multi-layer visual representations that arise in this model to those resulting from different training objectives on the same architecture.

Transfer learning approaches tend to be evaluated empirically, demonstrating through performance comparisons circumstances under which pretrained or transferred representations add value. While useful and pragmatic, this approach yields limited general insights into new problems. Instead of focusing on performance comparisons, we will focus on analysis of the representations at multiple layers of the same architecture trained with unsupervised, supervised, and reinforcement learning methods. To perform this analysis, we exploit tools that have been commonly applied in the interpretation of biological neural data. Our hope is to develop intuition for the properties of these different representations in order to support future efforts to develop training procedures.

## 1.1 RELATED WORK

Transfer learning in the supervised context has been widely applied, studied and reviewed, including within the narrower context of deep learning (Bengio, 2012). Thus, we limit the scope of our discussion to connections to the most relevant literature on transfer learning in the context of reinforcement learning, as well as literature focusing on the analysis of representations learned.

**Transfer learning for reinforcement learning**   Within RL specifically, transfer learning has been of interest since before the recent deep RL boom (Taylor & Stone, 2009; Lazaric, 2012). There have been diverse attempts to learn representations from experience that would support generalization across other tasks, including a focus on predicting future states or rewards (Rafols et al., 2005; Lehnert et al., 2020). Relevant to our present setting, Hill et al. (2019) found specifically that generalization is facilitated by egocentric observations. In addition, transfer learning for reinforcement learning has been one way to operationalize the broader and more natural problem of continual learning, wherein representations must be learned, transferred, reused, and adapted repeatedly over the lifetime of an agent (Hadsell et al., 2020).

Among more recent deep RL approaches, it is also possible to leverage multiple objectives for learning representations concurrently. That is, rather than first learning a representation from previously logged data and then learning a policy from that pretrained representation, it is possible to perform deep RL with auxiliary objectives (Jaderberg et al., 2017) including self-supervised tasks such as predicting past or future experience (Wayne et al., 2018; Schwarzer et al., 2021). Critically, concurrent learning with unsupervised or self-supervised objectives as well as RL offers the advantage that as RL proceeds and the data distribution changes, the auxiliary tasks continue to train on the shifting (and increasingly relevant) data.

As constrastive approaches have become popular, they have also been explored as auxiliary losses, such that policies rely on features shaped both by the contrastive and RL objectives (Oord et al., 2018; Laskin et al., 2020). While intuitively it seems reasonable that joint training of a representation with RL and an additional loss may help performance, new results by Stooke et al. (2021) keep alive the possibility that exclusively pre-training a representation from unsupervised objectives could outperform end-to-end RL. Nevertheless, it presently remains far from resolved which objectives used for pretraining or as auxiliary tasks (i.e. concurrent with RL) will work best. Indeed, in one of the larger empirical studies to date, Yang & Nachum (2021) sweep over many pre-training objectives. Their results "suggest that the ideal representation learning objective may depend on the nature of the downstream task, and no single objective appears to dominate generally" (with the caveat for our setting that their environments are relatively simple control environments and they focus on state rather than image observations).

**Analysis of neural representations**   In general, the present state of the literature on transfer of representations for RL leaves us with meaningful leads as to what objectives to try, but still incomplete insight into why various objectives add value or how to anticipate what will work well on a new problem. However, there is precedent both in the machine learning literature as well as biological neural analysis literature for attempting to analyze learned representations.

Taskonomy (Zamir et al., 2018) determined transfer learning performance across the same architecture trained with several supervised and unsupervised tasks. While this work did not compare representations directly, one relevant finding was that autoencoders tended to be an outlier in the revealed task structures. Furthermore, the Taskonomy networks were also used to predict fMRI activity (Wang et al., 2019), and it was found that several tasks related to 3D image processing made very similar predictions, while the autoencoder was not highly similar to any other models.

In Zhuang et al. (2021), several different unsupervised networks and one supervised network are assessed as models of primate visual processing. This work found that contrastive unsupervised methods generally had good transfer performance and predicted primate neural activity well. Note that neural predictivity is only an indirect way of comparing components of network representation; we perform direct comparisons across different networks, including an RL-trained model (which is to our knowledge the first time such a full comparison has been done).

**Neural analysis metrics**  We use representation similarity analysis (RSA) to compare representations across networks. RSA has been used extensively in neural network analysis and has connections to many other metrics used in neuroscience (Kriegeskorte & Wei, 2021). Historically, the concept of sparsity has also been important in neural analysis. Sparsity can be defined several different ways (Willmore & Tolhurst, 2001) but in general sparse representations have been associated with efficient encoding of natural stimuli statistics, effective associative memory, and enhanced downstream classification (Rolls & Treves, 1990; Olshausen & Field, 1996; Babadi & Sompolinsky, 2014; Olshausen & Field, 2004). Another commonly-measured property of neural representations is dimensionality, which is used as a way to probe the number of latent variables encoded in a neural population as well as understand how they are embedded (Jazayeri & Ostojic, 2021). Higher dimensionality is also associated with better downstream classification (Rigotti et al., 2013).

## 2 METHODS

### 2.1 MODELS AND TRAINING

Table 1 summarizes the models trained for this study (for all but RLRod, we train three instantiations of each). For more details on these networks and training see Appendix A. The ResNet architecture shared by all models is shown in Figure 1A. The number of feature channels is 16 for the first meta-layer and 32 thereafter. The final layer is a 128-unit fully connected layer.

The images used are all 64x64 RGB egocentric images generated from the virtual rodent environments Merel et al. (2020) built in MuJoCo (Todorov et al., 2012) using `dm_control` (Tunyasuvunakool et al., 2020). For the RL model, the agent creates its own training images through exploration in the environment (see Appendix A.0.1). Other models were trained using images generated by the agent's exploration – these images, along with some other features of the rodent's state and action output, have been made publicly available and documented for offline RL in Gulcehre et al. (2020).

Example images can be seen in Figure 1B. The rodent engages in one of four possible tasks at a time: bowl escape (where the rodent must crawl out of an uneven valley), gaps run (the rodent must run and jump over gaps), two-tap task (the rodent must tap an orb twice with a set amount of time in between), and maze forage (the rodent has to find orbs in a maze structure).

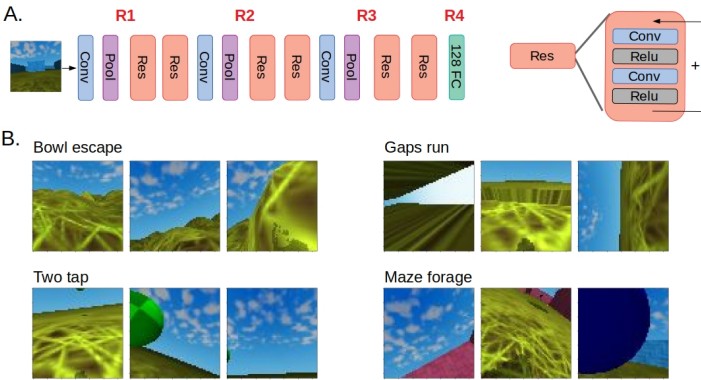

Figure 1: The network architecture (A) and images used for training (B), which were drawn equally from the four different virtual rodent tasks. The layers from which activity was recorded for analysis are labeled in A. For R1, 2 and 3 the Relu layer after the Pooling was recorded.

| Training style | Name (Abrv.) | Description | Output Dimension |
|---|---|---|---|
| Supervised | Reward model (SupRwd) | Predict reward received after this frame | 1 |
| Supervised | Task model (SupTsk) | Classify which task the rodent is in | 4 |
| Supervised | Orientation Model (SupOri) | Predict egocentric orientation in space | 3 |
| Supervised | All-Task Model (SupAll) | Do all of the above tasks | 8 |
| Unsupervised | Autoencoder (UnsAut) | Reconstruct the image | 64x64x3 |
| Unsupervised | Variational Autoencoder (UnsVar) | Reconstruct the image | 64x64x3 |
| Unsupervised | Contrastive Predictive Coding Model (UnsCpc) | Classify pairs of images as consecutive or not | 1 |
| Untrained | Random (Rand.) | Random weight initialization | - |
| Reinforcement | Rodent Model (RLRod) | Achieve high reward on 4 embodied tasks | 38 |

Table 1: Description of models compared. For details, see Appendix A

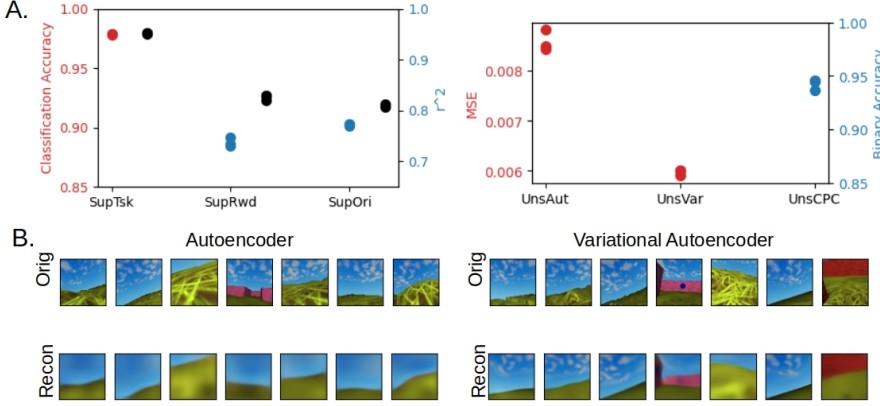

Figure 2: Performance of trained networks: supervised models in A, left (classification accuracy for the task model; $r^2$ values for the reward and orientation models) and unsupervised models in A, right (mean squared error for image reconstructions for the standard and variational autoencoders are shown; binary accuracy for the contrastive predictive coding model.) Each point represents one of three random instantiations. Chance for task classification is .25 and for CPC is .5. For SupOri, $r^2$ is averaged over all three dimensions. Performance for the SupAll networks is shown in black for each task. Example reconstructions from the two types of autoencoders are shown in B.

## 2.2 ANALYSIS METHODS

Activity was recorded from four different layers (marked as R1-4 in Figure 1A) in response to 2048 test images drawn equally from the four tasks. Here we briefly describe the analyses applied, with more detailed descriptions available in Appendix A.

**Representational Similarity Analysis** RSA was used to determine the extent to which different models represent visual inputs similarly. First, dissimilarity matrices were made for each layer in each network (see Eqn 2; example dissimilarity matrices in Figure 4A). RSA matrices (resulting from two different correlation metrics) indicate how similar these dissimilarity matrices are across networks. Separately we also performed RSA across layers within networks and included pixel and action spaces.

**Sparsity** To measure the sparsity of representations in these networks we use a lifetime sparsity metric which determines the extent to which a neuron responds selectively to different images (Vinje & Gallant, 2000).

$$s = \frac{1 - (\frac{1}{n})(\frac{(\sum r_i)^2}{\sum r_i^2})}{1 - \frac{1}{n}} \qquad (1)$$

where $n$ is the number of images and $r_i$ is the response of the neuron to image $i$. A sparsity value of 1 indicates very selective responses, with 0 indicating equal responses to all inputs.

**Dimensionality** We perform PCA on population activity and then estimate embedding dimensionality using both the participation ratio (see Eqn 3) and the number of PCs needed to reach 85% variance explained.

## 3 RESULTS

### 3.1 MODEL PERFORMANCE

We show the test performance of our supervised and unsupervised models here (for more information on the performance of the RL agent see Merel et al. (2020)).

All models reach an acceptable level of performance, though some are clearly better than others, which may be relevant for how to interpret their different representations. For example, the variational autoencoder performs much better than the standard autoencoder as can be seen both in the MSE measure and in the example reconstructions shown in Figure 2B. The joint training of SupAll also seems to provide a performance boost compared to the individually trained networks.

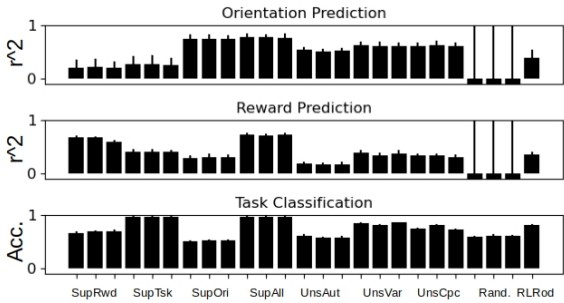

Figure 3: Transfer learning performance using final layer (R4) activity for supervised, unsupervised, random, and RL networks.

### 3.2 TRANSFER LEARNING PERFORMANCE

To understand what kind of information is present in the outputs of each trained network (layer R4 in Figure 1A) and to compare to previous transfer learning work, we trained classifiers (linear or logistic regression) to readout the three variables used for supervised training: task, reward, and orientation. The test performance of these classifiers (across 20 test-train splits) is shown in Figure 3.

As expected, the supervised networks perform well on the tasks they were directly trained on. Many of the networks also perform well on task classification, including the random untrained ones. UnsVar and UnsCpc perform fairly well on all three tasks as well. The RL model performs well on task classification and relatively well on orientation and reward prediction. These results align with previous claims about the generalizability of certain unsupervised representations and adds that RL representations may generalize as well.

### 3.3 REPRESENTATIONAL SIMILARITY ANALYSES

The RSA matrices in Figure 4 compare the representations of the different networks at each layer individually.

First, we can see there is little variability across different instantiations of networks trained the same way (this can be seen as the large 3x3 blocks on the diagonal). Another strong finding is that networks trained with different algorithms become less similar across the layers in the network. Specifically, in R4, many similarity values are near 0, whereas in R1 many are closer to 1 (this can be seen most clearly in the summary figure provided in Figure 4B). This may be a result of the loss function having stronger influence at later layers.

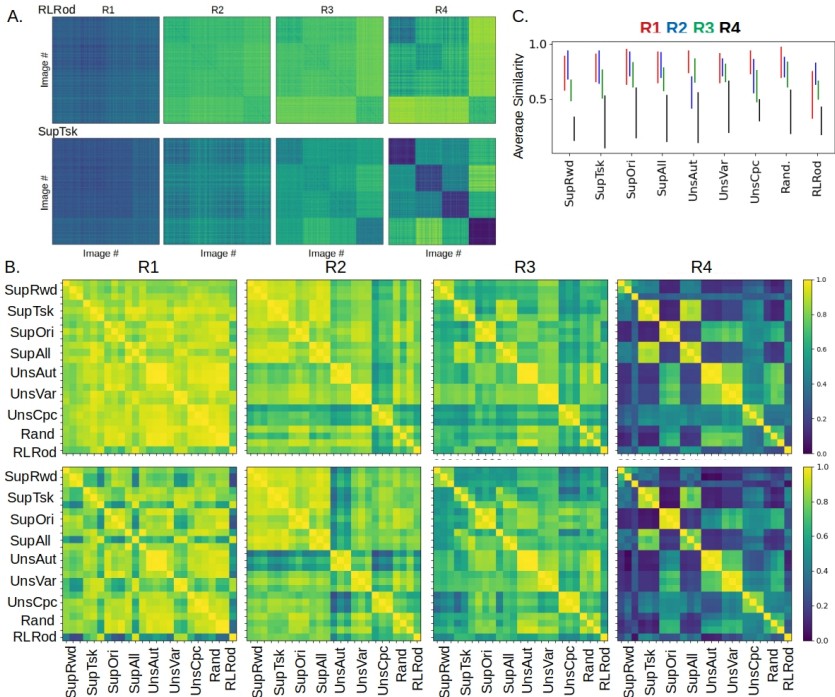

Figure 4: A. Example dissimilarity matrices for the four recorded layers (R1-4) indicating the extent to which a pair of images is represented similarly or not (higher values are less similar). Images are grouped according to task type. Color range: 0-1.2 Dissimilarity matrices for all networks can be seen in Supp. Fig 9 B. The correlation between these dissimilarity matrices for pairs of networks is given in the RSA matrices. On the top (bottom), we use the whitened cosine metric (Spearman's rho rank correlation) to measure this correlation. Higher values are more similar. C. Summary of the Spearman RSA results. Each line shows the mean ± standard deviation of how similar the network type is to all other network types, for each layer.

Comparing representations across networks, we see in Figure 4C that on average the RLRod model is the least similar to the others, particularly at layers R1-3. Looking at what models do have similarity to RLRod, at layers R1, R3, and R4, the SupTsk networks show some of the best matches to RLRod (followed mainly by SupAll). This suggests that identifying, at least implicitly, which of the four possible tasks the rodent is in is part of what the RL agent has learned to use its visual encoder for. This aligns with the high performance on task classification shown by RLRod in Figure 3.

Interestingly, throughout the layers the random (untrained) networks have fairly high similarity to the trained networks. This suggests that the architecture alone has a large influence over the representations generated by these networks in response to these particular images. While for many of the trained networks, this similarity fades by R4, it is still somewhat high for the UnsAut, UnsVar and SupOri networks.

We also created RSA matrices that represent how information is transformed within a single network. Here, in addition to using layer activity we also compare activity to pixel space and to the 38-dimensional action space of the rodent, which is the output of the full rodent agent model (see dissimilarity matrices for these spaces in Supp. Fig. 11). Examples of these matrices are shown in Figure 5A, with all shown in Supp. Fig. 10. Looking at the RSA matrix for the RLRod network, we can see this network fades somewhat gradually from representing pixel space to starting to represent action space, whereas other networks, like the untrained ones, retain a strong similarity to pixel space. In some networks, such as RLRod and UnsVar, R2 is more similar to pixel space than R1. This odd effect seems to be the result of the maze task being quite distinct in pixel space, but this distinction doesn't emerge until later in trained networks (see Supp Figs 9 and 11).

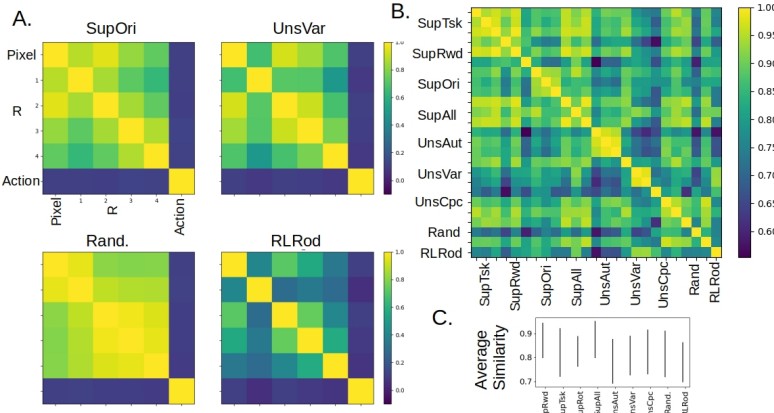

Figure 5: How networks transform their representations across layers. Here we do RSA on the activity of all recorded layers (R1-4) along with pixel inputs and the 38-dimension action space generated by the RL agent. In A, example RSA matrices are shown for different networks. In B, the Spearman rank correlation between these RSA matrices is shown. In C is a summary of the similarities in B, as in Figure 4C.

Looking at the comparison across all networks in Figure 5B, we find a bit more diversity within networks trained the same way than suggested by Figure 4. However, here still the RLRod network is on average the least similar to the other networks (with UnsAut a close second), as can be seen in Figure 5C.

### 3.4    SPARSENESS

Noting that the RLRod network is different from the others, we wanted to look further into how its representations differ. We started by looking more closely at the dissimilarity matrices for the RLRod model. Figure 4A shows the dissimilarity matrices for RLRod and for one of the best matches to RLRod, the SupTsk network. Of note is that starting at layer R2 the RLRod network has much higher dissimilarity values than SupTsk (and indeed higher than other networks, as can be seen in Supp Fig 9). That is, pairs of images that are represented with somewhat similar patterns of neural activity in the SupTsk network have very disparate representations in the RLRod network. Indeed, the correlation between activity patterns is near zero or even negative for most pairs of images in the RLRod model.

One possible way of creating such dissimilarity is to have a sparse representation wherein individual units are only responsive to a small and specific subset of images. In the extreme, having each neuron respond to one and only one image would make the responses to any pair of images highly dissimilar. As can be seen in Figure 7, the RLRod representations are indeed sparser than the other networks, particularly at layers R3 and R4. This suggests that reinforcement learning may naturally generate sparse neural activity as a way of differentiating different visual states.

### 3.5    DIMENSIONALITY

Another metric commonly used in the neuroscience literature is that of the embedding dimension (Jazayeri & Ostojic, 2021). Embedding dimension is commonly estimated using linear dimensionality reduction. Here, we show the estimated embedding dimensions according to two metrics (see Methods A.0.4) for all layers and networks in Figure 7.

One striking finding is the high dimensionality of the RLRod model, particularly past layer R1. Dimensionality of a neural representation can also be related to sparsity in that an extremely sparse population wherein there is a one-to-one correspondence between images and neurons will have dimensionality determined by the number of images. In this way, these results align with the high sparsity of the RLRod model.

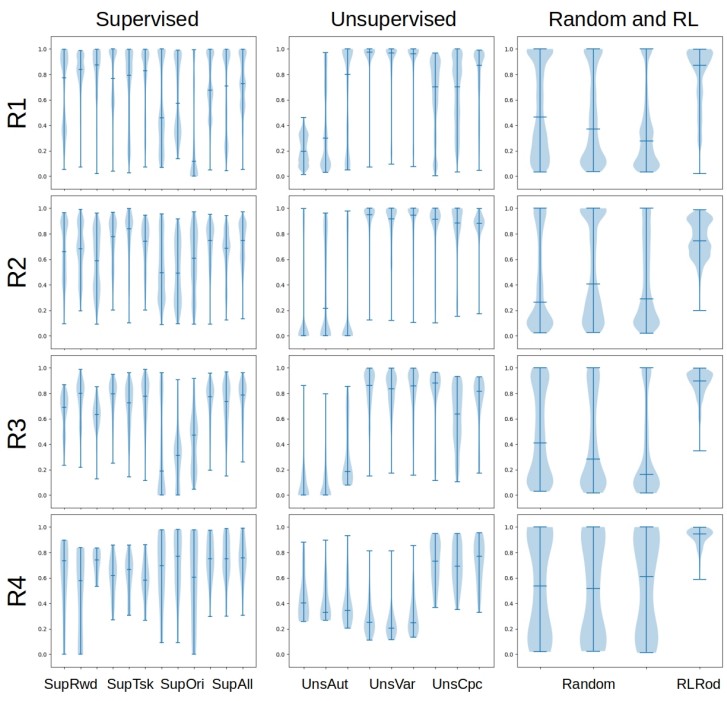

Figure 6: Sparsity distributions across layers and networks (median values marked). Higher sparsity values means individual units respond selectively to a smaller number of images.

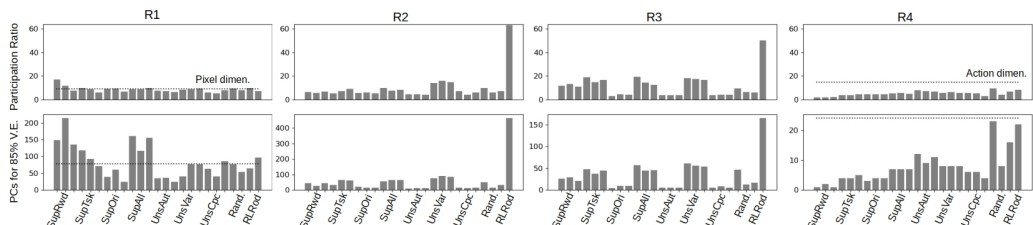

Figure 7: Dimensionality of representations. Top row shows participation ratio at each layer for all networks (same y-axis across layers); Bottom row shows the average number of principal components needed to reach 85% variance explained. In the R1 (R4) plot, the dimensionality of the pixel space (action space) is indicated with a black dashed line.

Looking at the number of PCs needed to explain 85% variance at R4, the supervised models have average dimensionality roughly equivalent to the dimensionality of the tasks they were trained to perform (SupRwd, 1.33; SupTsk, 4.33; SupOri, 3.67; SupAll, 7). Extrapolating to the other models we can say that UnsCPC model is using 5.33 dimensions to perform its binary classification; UnsAut and UnsVar are using 10.67 and 8 dimensions respectively for their reproductions; and the RLRod visual encoder is performing a 22-dimensional task. Such a high dimensional representation may be crucial for the RL model to perform high-dimensional control over a range of environments and timescales. The high dimensionality of the untrained networks at R4 suggests that low dimensionality is learned by the supervised and unsupervised networks. Ways to retain high dimensionality through supervised or unsupervised training may thus make them more beneficial as pre-training for RL.

## 3.6 SUPERVISED TRAINING OF ACTION SELECTION

It is conceivable that the high dimensionality of RLRod is due to its use in predicting the 38-dimensional action space (whose dimensionality according to these metrics is indicated by the black

dashed line in Figure 8). To test this, we trained models to predict actions (i.e., behavioral cloning). Interestingly, the visual encoder alone performs very poorly on this task (Figure 8) implying that proprioception is necessary for reliable action modeling. We thus tried to add a portion of the original RL architecture by training a parallel MLP that encodes proprioceptive information. Proprio- and visual network outputs are concatenated, fed into a 128-unit layer then used jointly to predict action. This network performs much better (Figure 8; predicted versus real values for each output dimension can be seen in Supp. Fig. 12), but surprisingly does not use the visual inputs at all for this performance. In fact, it learns to zero-out the activity in the visual encoder, and performance while zero-ing visual inputs to the network is nearly identical. The fact that proprioceptive information alone is sufficient to reach this level of action prediction is also supported by the performance of a network trained with only the proprio-MLP.

Note that vision is necessary both for task identification and good performance on these RL tasks due to their design. What these results suggest is that RLRod is not using visual inputs to control immediate actions, as single-step action selection cannot be done with vision. However, proprioceptive information alone still only achieves $r^2$ of around .4. Therefore visual information is still playing a role, but likely on longer timescales: via interaction with the downstream LSTMs visual input is used to identify goals and plan longer term actions .

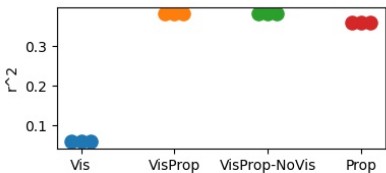

Figure 8: Action prediction performance (averaged over all 38 dimensions) for a network trained with just a visual encoder, visual and proprioceptive encoder, visual and proprioceptive encoder but visual inputs set to 0 during test, and a proprioceptive network alone (this network predicts actions immediately from the output of the 3-layer proprio MLP). Three networks are shown for each but performance is overlapping. Performance broken down by action dimension in Supp. Fig 12.

## 4 DISCUSSION

In this work, we've sought to systematically characterize the population level properties of the neural representations that emerge when training neural networks on egocentric visual observations with different objectives. While some works actually do provide a few anecdotal visualizations of the image features emphasized by RL as well as unsupervised objectives (e.g., Stooke et al., 2021), we are not aware of efforts to more comprehensively compare the properties of representations learned by RL with those that arise from other objectives. To this end, we have leveraged analysis techniques adapted from neuroscience, which have rather consistently shown the RL-derived representation to be an outlier. In particular we find that representations become less similar across networks at later layers, unsupervised objectives can capture some features of the RL-trained network such as sparsity and high dimensionality at certain layers but no single method captures all RL features, and the RL-trained visual encoder seems to drive long-term behavioral planning.

We want to clarify that representations which differ from those that arise from end-to-end RL may be useful for pre- or joint-training of some tasks in some environments. But insofar as the representations differ in critical ways, enriched understanding of the properties of the different resulting representations may help us better anticipate which objectives support transfer in which settings and why. For example, high representational dimensionality and representational diversity across inputs may serve as good stand-ins for how easy it will be to learn controllers for a wide range of tasks.

Finally, these results may also be of interest to the neuroscience community, which has studied sparsity of neural representations for decades. While simple learning procedures have been shown to create sparsity (Földiak, 1990), the fact that it arises here from reinforcement learning may further contextualize its origins and role in the brain.

REPRODUCIBLITY STATEMENT

Here we used images openly available through RL Unplugged (Gulcehre et al., 2020). Code for analysis and training of the supervised and unsupervised networks is included in uploaded supplementary materials.

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

## APPENDIX

## A   EXTENDED METHODS

In all but the RL model, networks were trained end-to-end with ADAM using batch size of 128, L2 weight regularization of .001 and learning rate of .001 (dropping three times during training). These networks were trained for 75 epochs (this value was chosen through preliminary explorations that indicated validation curves tended to plateau around this stage). Three different random instantiations of each type of network were trained.

### A.0.1   REINFORCEMENT LEARNING MODEL

For an example of a ResNet trained through reinforcement learning on a complex visual task, we use the visual encoder from the virtual rodent model introduced in Merel et al. (2020). This agent takes in egocentric visual and proprioceptive information to control a realistic rodent-like body to perform the four above-mentioned tasks. After the visual encoder, visual outputs are combined with encoded proprioceptive information, which then feeds into a recurrent neural network that is trained to estimate value. These networks also feed into a recurrent policy network, which defines the movement of the rodent body in a 38-dimensional continuous space. Rewards for the maze and two-tap tasks are sparse upon orb hit, whereas bowl escape and gaps have denser rewards. Here we use weights only from the visual encoder of the trained model, provided by the authors of Merel et al. (2020).

### A.0.2   SUPERVISED, UNSUPERVISED, AND RANDOM MODELS

While the images generated from the agent's actions don't have obvious labels, there is some associated information that can be used for supervised learning. Here, we train four different models: a task model (SupTsk), a reward model (SupRwd), an orientation model (SupOri), and a model that performs all three tasks (SupAll). Weights for supervised models were initialized with a truncated normal distribution with standard deviation .05.

**Task Model:** In SupTsk, the final 128 unit layer of the ResNet is followed by a 4 unit layer trained with categorical cross entropy loss to classify an image as coming from one of the 4 different tasks described above that the RL agent was trained to perform. Because the RL agent is not given an explicit cue about which task it is in, it must deduce it from visual information; therefore we can assume that task identity is determinable from visual inputs.

**Reward Model:** In SupRwd, the final layer of the ResNet is followed by a single sigmoidal unit which is trained with a mean squared error loss to predict the reward the agent received after the action it took in response to the image (the performance of this model is therefore constrained by the extent to which a single frame is sufficient to predict upcoming reward). To make the distribution of rewards more similar across tasks, all tasks were normalized to have max reward of 1 and the sparse

rewards of the maze and two-tap tasks were made dense by assigning small reward to several frames leading up to the reward.

**Orientation Model:** In SupOri, the final ResNet layer is followed by a 3 unit tanh layer that is trained with mean squared error loss to predict the rodent's body position in 3-D egocentric space (i.e., equivalent to the yaw, roll, and pitch).

**All Task Model:** In SupAll, three parallel layers are added after the 128-unit layer that perform each of the above tasks.

For unsupervised learning, we trained an autoencoder (UnsAut), a variational autoencoder (UnsVar), and a contrastive predictive coding model (UnsCPC). Unsupervised network weights were initialized with the Glorot uniform distribution Glorot & Bengio (2010).

**Autoencoder:** For UnsAut, the ResNet architecture is rebuilt in reverse after the 128 unit layer. It is trained on image construction using the binary cross entropy loss. (i.e., undercomplete autoencoder; Goodfellow et al. (2016))

**Variational Autoencoder:** For UnsVar, after the final 128-unit layer, mean and variance layers are created which feed into a sampling layer followed by the reverse ResNet architecture and the network is trained using a reconstruction loss plus KL divergence loss (Kingma & Welling, 2013) (we used code provided by Chollet (2020)).

**Contrastive Predictive Coding Model:** For UnsCpc, we implement the architecture and learning procedure from Oord et al. (2018). Briefly, the encoding network is trained to predict whether two images are of consecutive frames from the rodent's exploration or not. We use code adapted from Tellez (2018).

**Random Model:** To assess the impact of architecture alone on representations we include untrained or random (Rand.) models. These were initialized using the Glorot uniform distribution.

### A.0.3 REPRESENTATIONAL SIMILARITY ANALYSES

Representational similarity analysis (RSA) is a two-step analysis used to determine the extent to which different models represent visual inputs similarly. First, dissimilarity matrices were made for each layer in each network. Each entry in the dissimilarity matrix is defined as

$$d_{ij} = 1 - corr(\boldsymbol{r}_i, \boldsymbol{r}_j) \tag{2}$$

where $\boldsymbol{r}_i$ is the population activity in response to image $i$ and $corr$ is the Pearson correlation coefficient.

Example dissimilarity matrices from two networks can be see in Figure 4A.

These dissimilarity matrices are then compared to determine how similar two networks are. We use the RSA toolbox (Nili et al., 2014) to perform these comparisons. Specifically, we use two metrics to ensure our findings are robust: the whitened cosine metric, which is equivalent to linear centered kernel alignment, and Spearman's rho rank correlation.

We also use RSA to compare different layer representations within a network. For this we create the same dissimilarity matrices for layer activity, and also for pixel values of the input images and the 38-dimensional output space of the virtual rodent. These dissimilarity matrices were compared using the Pearson correlation coefficient to create RSA matrices. These RSA matrices were then compared across networks using the Spearman rank correlation.

### A.0.4 DIMENSIONALITY REDUCTION

Embedding dimension is defined as the number of dimensions (in ambient Euclidean space) explored by the neural activity (Jazayeri & Ostojic, 2021). To estimate the embedding dimension of the activity at different layers in these networks, we perform principal components analysis (PCA) on the population responses. We then calculate the participation ratio:

$$PR = \frac{(\sum \mu_i)^2}{\sum \mu_i^2} \tag{3}$$

where $\mu_i$ is the eigenvalue associated with the $i$th PC. This value can correspond to the number of PCs needed to reach 80-90% variance explained under certain distributions of the eigenvalues of the covariance matrix (Gao et al., 2017). As a secondary measure we also calculate empirically the number of PCs needed to reach 85% variance explained.

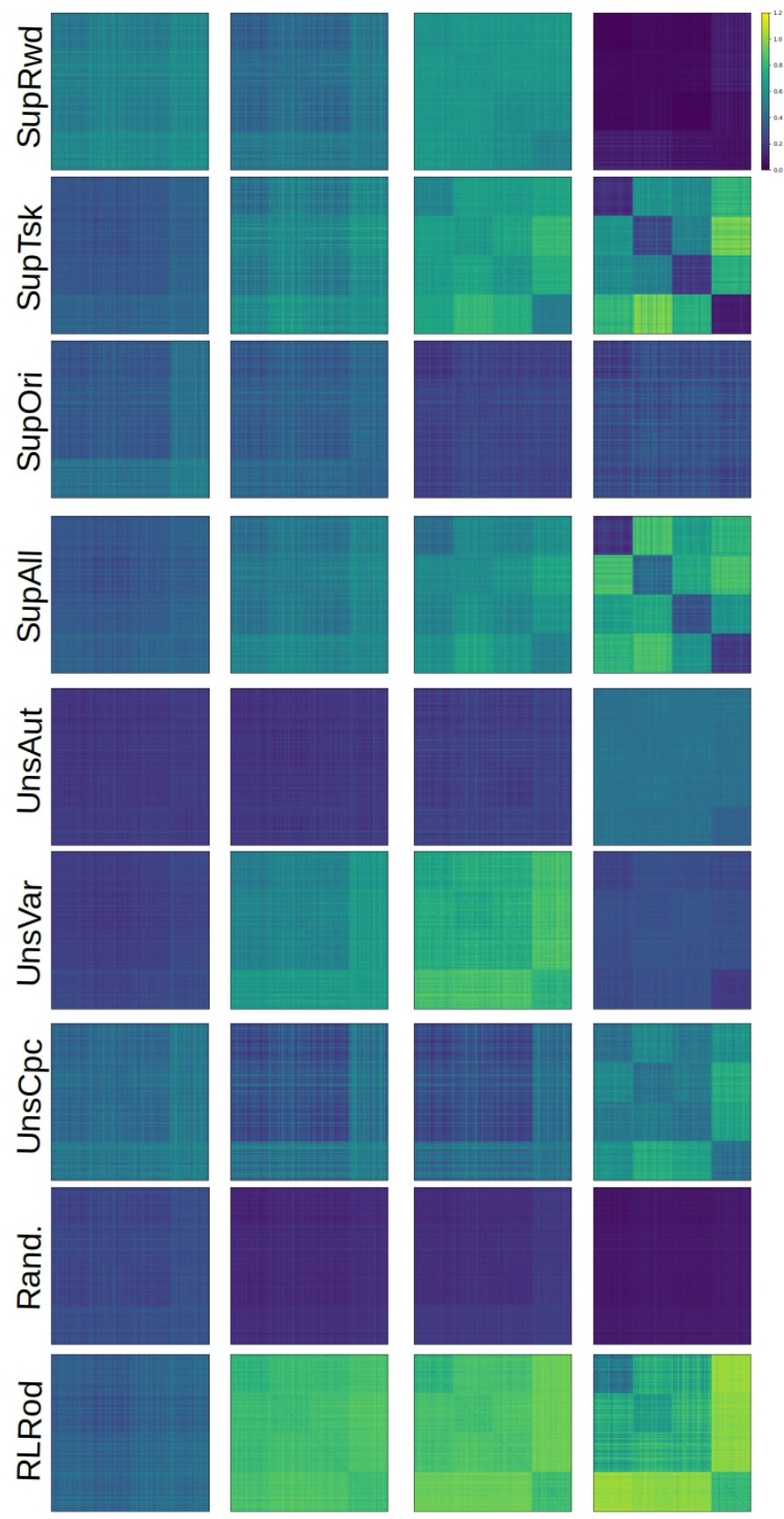

Figure 9: Supplementary Figure: Dissimilarity matrices for all networks. Color range: 0-1.2

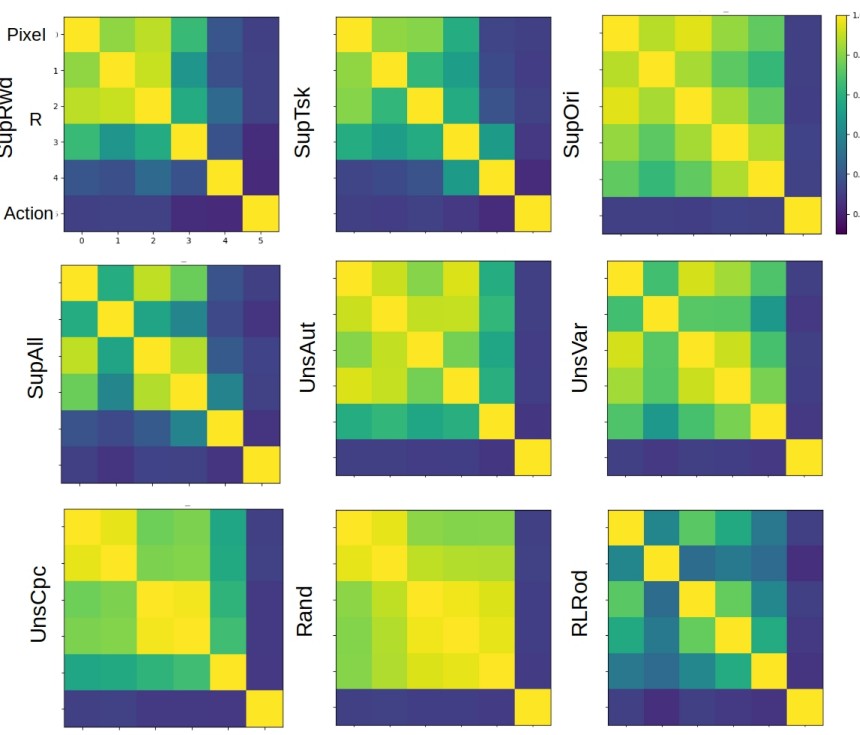

Figure 10: Supplementary Figure: Within Network RSA matrices for all networks.

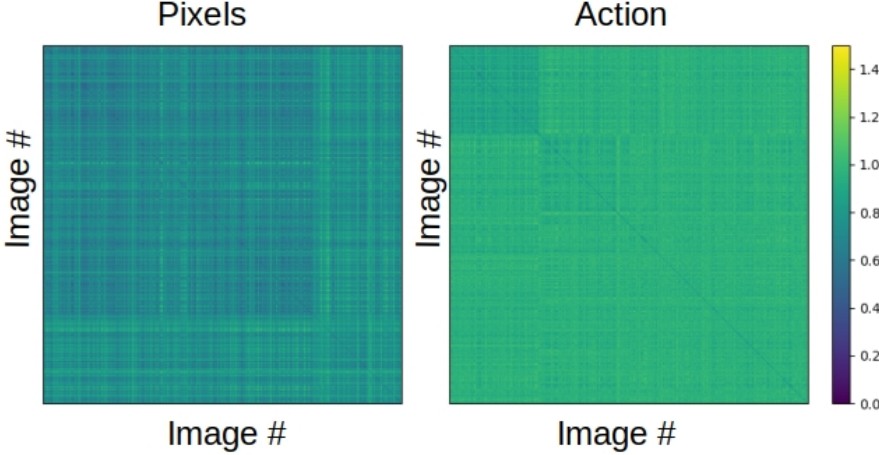

Figure 11: Supplementary Figure: Dissimilarity Matrices for pixel and action spaces (note these are on a different color scale than those in Figure 9

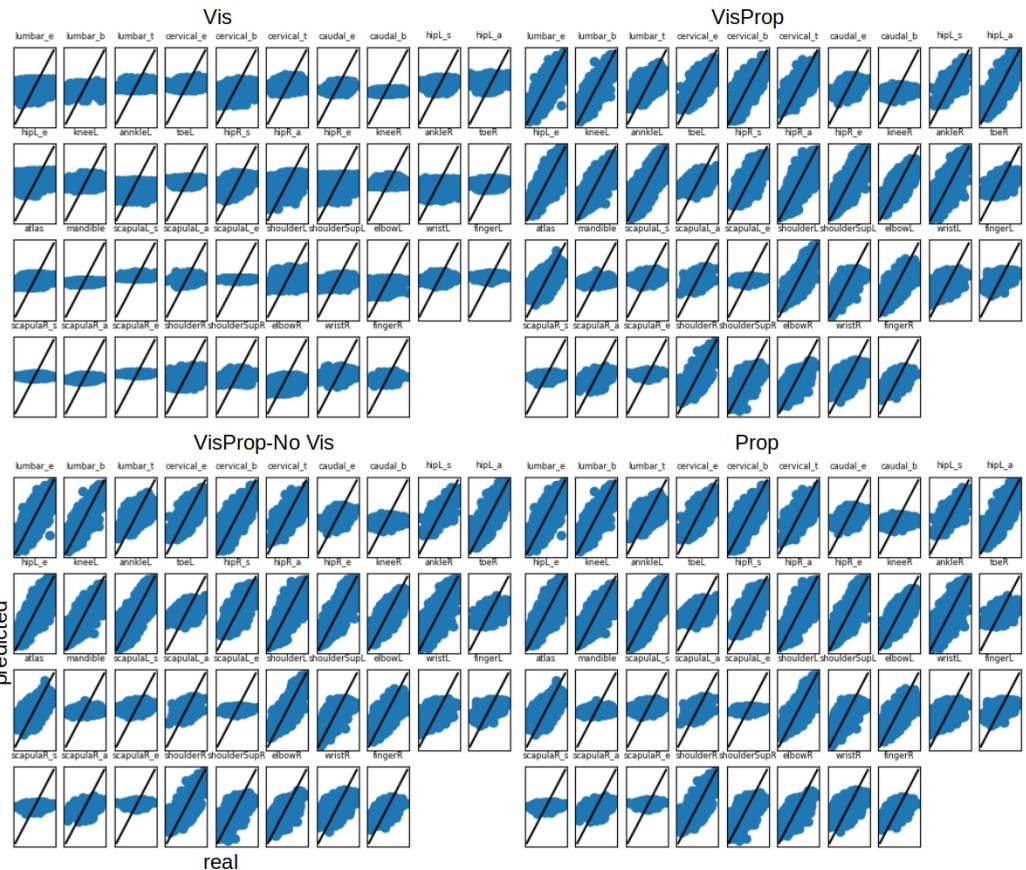

Figure 12: Action prediction from different supervised networks, as labeled in Figure 8. b = bend, t = twist, e = extend, a = abduct, s = supinate

