# OpenReview forum: "Divergent representations of ethological visual inputs emerge from supervised, unsupervised, and reinforcement learning"
_ICLR.cc/2022/Conference — ICLR 2022 Submitted_

### Official Review · Reviewer_WoDx · 2021-11-01

**Correctness:** 3
**Technical Novelty And Significance:** 3
**Empirical Novelty And Significance:** 4
**Recommendation:** 6
**Confidence:** 3

**Main Review:**

pros:
* comparing representations of networks trained with different objectives is a very timely question, and I am frankly really glad someone is working on this direction. Results here will inform not just efficient models in reinforcement learning, but, as the authors also point out, neuroscience as well (which is where I'm coming from).
* the setup is well done: the same architecture is used for all trainings, with 3 different variants per objective, and everything is trained in the same environment, i.e. very similar/the same images.
* figures generally well support the claims, and comparisons are done thoroughly with additional investigation into the underlying reasons for the RL-trained model's dissimilarity to other models

cons:
* simplistic setting: all models are trained in a rather simple environment without much variation in the images (Figure 1). Because of that, supervised models are not trained in the more common object classification paradigm, but instead to e.g. predict reward and classify tasks. This makes me worried that results might not generalize to more challenging environments and classification tasks
* potential confounds to determine similarity between objectives: what if RLRod simply has not learned the task well enough? I.e. if you were to continue training, its representations would become more similar to the representations learned by other objectives? Better establishing the "comparability" of networks at the stages of training they were captured at would be good here. For instance, in Figure 3, the RLRod model is the one that does not perform as well, so I wonder if its representations just have not converged. This could also explain the high dimensionality (lack of invariance) in Figure 7
* figures are difficult to read: while I appreciate the showing of all 3 models and all sub-tasks within a category (e.g. SupRwd, SupTsk, SupOri, SupAll for supervised), it does make the figures difficult to interpret, especially without additional aids such as different colors and symbols. Perhaps you could add condensed plots next to the more detailed versions that average over the 3 models or even the sub-tasks?

minor:
* When computing similarity to other models, how is each condition weighted? Are supervised approaches just "more similar" because there are more supervised models compared to e.g. the *single* RLRod network?
* I didn't get how "neural predictivity is only an indirect way of comparing components of network representation" -- it evaluates one aspect of representations, just like RSAs evaluate a different aspect. None of them are complete. Neural predictivity has an additional linear mapping step, RSA has an additional image-wise similarity step. https://arxiv.org/abs/1905.00414 is a useful paper discussing the differences
* order between SupRwd and SupTsk is different in table 1 and figure 2A
* the influence of architecture on representations (2nd-to-last paragraph page 6) has recently been the attention of some related work in neuroscience as well, e.g. https://www.biorxiv.org/content/10.1101/2020.06.08.140111v1 and https://openreview.net/forum?id=rkxcXmtUUS
* typos: figure 4 caption is missing a "." before "B."; page 7 "As can be seen in Figure 7" should be 6 I believe; and page 9 "dashed line in Figure 8" should be 7

**Summary Of The Paper:**

The paper explores how similar the representations learned by supervised, unsupervised, and RL techniques are in an egocentric virtual rodent environment. The main claim is that representations in RL-trained networks are most different from supervised and unsupervised networks, which is tested with an RSA metric on 4 layers of the networks. This is explored further by analyzing the sparsity, dimensionality (RL has much higher dimensionality) and action prediction (proprioception alone without visual features is sufficient here).

**Summary Of The Review:**

This paper has definitely made me think about some questions in the similarity of representations from different objectives, so I am inclined to accept. However, it also has limited applicability due to the simple environment and objectives, and could use some improvements in presenting a more general finding rather than detailed individual models (see main review for suggestions).

---

### Official Review · Reviewer_RDPx · 2021-11-02

**Correctness:** 3
**Technical Novelty And Significance:** 3
**Empirical Novelty And Significance:** 2
**Recommendation:** 3
**Confidence:** 4

**Main Review:**

Strengths:
While I am not particularly familiar with this subarea, they claim that this is the first time that an RL network is compared with the other types, so there is some novelty.
Also, the comparisons are fairly thorough, and the methodology is sound.
The results are fairly clear, if not totally unexpected. For example, they analyze different modules of the resnet backbone and find similarity between layers decreases with depth. This makes sense, as later layers will be more task oriented (like an information bottleneck analysis).

Weaknesses, with concrete, actionable feedback
A main weakness is that the results are pretty much what one might expect. For example, the RL network has to decide between 38 different actions, while the outputs of the other networks are low dimensional classification or regression problems. Thus it is not surprising that it creates high dimensional representations.

Another result is that networks trained the same way have similar representations. Another is (as mentioned above) that in networks trained for different tasks, the similarity between their layers decreases the deeper the layer.

Another result is that the RL network transfers fairly well to the objective of predicting the task, which would also be expected.

So, all of these results are not particularly surprising.

In the cases where the results are not completely expected, the authors remark on them, but do not perform any additional analysis to see why these results held. For example, for the RL network and the variational autoencoder, the second module's representations are more similar to pixel space than the first module's. I.e., the representation diverges from pixel space in the first layer, and then becomes more similar in the next layer up. They try to explain this in one sentence as having to do with the maze task (which I don't find convincing). This doesn't explain why the variational autoencoder would do this. This is one result that perhaps could be analyzed more closely as to why it happens.

Another somewhat puzzling result that isn't remarked on is in Figure 7, where the number of principal components needed to explain 85% of the variance within representations over the dataset is measured. This gives a measure of the dimensionality of the representation. What I find puzzling is that the dimensionality of, for example, the random network makes a very large jump between the 3rd module and the 4th. Maybe I'm just bad at math, but it is unclear to me how the layers' dimensionality jumps by about 4X between these two layers. However, it does seem to be a property of many of the networks, that the intrinsic dimensionality increases gradually from layer 2 to layer 4. Why is this?

In order to test whether it is the 38-dimensional output of the RL network that causes the high dimensionality, they train a network to predict the actions from the input images (behavioral cloning). This doesn't work well, but adding proprioceptive input does succeed (what proprioceptive inputs are here should be explained for those of us not familiar with the rat RL task). But then they don't measure the dimensionality of the representation, which I thought was the point of doing this. There is also the puzzling result that the network basically turns off the image input in favor of the proprioceptive input. This makes sense if the proprioception indicates what the rat is about to do - e.g., if there is the equivalent of a muscle tension proprioception, this certainly could be indicative of the actions.

Minor points/typos:
Figure 2A is not referred to in the text. It should be in the paragraph under section 3.1.

There are several places where the text refers to the wrong Figure - e.g., mixing up Figures 7 and 8.

Page 6, second paragraph, you might mention the deep image prior paper here. (https://arxiv.org/abs/1711.10925).
Page 6, third paragraph, you might mention the information bottleneck here.

Section 3.4, second paragraph, 3rd line from the bottom, it should be Figure 6, not 7, I believe.

Top of page 9: this should be Figure 7, not 8.

**Summary Of The Paper:**

This paper compares the representations learned by otherwise identical networks (up to the output layer) for different tasks: supervised (4 tasks), unsupervised (autoencoders, vanilla and variational), self-supervised predictive coding, untrained (randomly initialized), and one RL policy network. They are all trained on the same images from the RL task, and the supervised tasks are related to the RL task (e.g., what task is this image from).

The representations are compared by using relatively standard neuroscience measures: RSA (with two distance measures), a kind of meta-RSA that compares the correlations between the RSA matrices of each network to each other (again with two correlation measures), two sparsity measures, and two measures of dimensionality of the representations.

They find that the RL network stands alone, treating images in complex ways, being very uncorrelated to the other networks, being more sparse, and more high dimensional. The unsupervised networks also stand out, although not as much, from the rest.

They also investigate transfer learning between the networks, training single-layer networks from the output of the penultimate layer of each network (frozen) on each other network's tasks.

I have read the authors' response, and given that they are not responding to individual reviews or changing the paper, I am keeping my score the same.

**Summary Of The Review:**

This paper does a nice job of comparing representations learned by an identical network architecture, given different training objectives. The results are interesting, but not completely unexpected. The idea of comparing an RL-trained network to other networks doing image classification and regression is a good idea, but the results are about what one would expect. Where they are not what one would expect, they are not explained or analyzed in any depth. Hence I don't believe this paper is ICLR-worthy.

---

### Official Review · Reviewer_mWVA · 2021-11-02

**Correctness:** 4
**Technical Novelty And Significance:** 1
**Empirical Novelty And Significance:** 2
**Recommendation:** 3
**Confidence:** 4

**Main Review:**

strengths:

- A thorough and novel characterization of various properties of representations learned via various training objectives

weaknesses:

- Logic of motivation for the work unclear. It's not clear what question this characterization helps to answer or why this analysis is most appropriate. Hypotheses are not clearly described nor are the impacts or take-home messages.
- Poorly written in places.
- Related work is missing and irrelevant work is cited. Similarity analysis, as well as analyses of dimensionality and sparsity are common in machine learning research, not just in neuroscience. It's not clear why work on neural predictivity is relevant here. Broad swaths of research are cited rather than a targeted background related to a specific question.
    - **Morcos AS**, **Raghu M**, **Bengio S**. Insights on representational similarity in neural networks with canonical correlation. .
    - **Leavitt ML**, **Morcos AS**. Selectivity considered harmful : evaluating the causal impact of class selectivity in DNNs. In: *International Conference for Learning Representations (ICLR)*. 2021.
    - **Leavitt ML**, **Morcos AS**. Linking average- and worst-case perturbation robustness via class selectivity and dimensionality. In: *arXiv:2010.07693*. 2021.
    - **Morcos AS**, **Barrett DGT**, **Rabinowitz NC**, **Botvinick M**. On the Importance of Single Directions for Generalization. In: *International Conference for Learning Representations (ICLR)*. 2018.
    - **Kornblith S**, **Norouzi M**, **Lee H**, **Hinton G**. Similarity of Neural Network Representations Revisited. In: *ICML*. 2019.
    - **Thompson JAF**, **Yoshua Bengio**, **Schönwiesner M**. The effect of task and training on intermediate representations in convolutional neural networks revealed with modified RV similarity analysis. In: *Cognitive Computational Neuroscience*. 2019.
    - **Raghu M**, **Gilmer J**, **Yosinski J**, **Sohl-Dickstein J**. SVCCA: Singular Vector Canonical Correlation Analysis for Deep Understanding and Improvement. .
    - **Ansuini A**, **Laio A**, **Macke JH**, **Zoccolan D**. Intrinsic dimension of data representations in deep neural networks. In: *Advances in Neural Information Processing Systems*. 2019.
    - **Recanatesi S**, **Farrell M**, **Advani M**, **Moore T**, **Lajoie G**, **Shea-Brown E**. Dimensionality compression and expansion in Deep Neural Networks . http://arxiv.org/abs/1906.00443.

**Summary Of The Paper:**

The paper includes a comparison of the representations learned according to a variety of training objectives, including reinforcement, supervised and unsupervised objectives, in a common architecture and with similar training data. The comparison looks at dimensionality, sparsity and representational similarity and finds representations learned via RL to be outliers relative to the other kinds of objectives.

**Summary Of The Review:**

The paper presents several interesting characterizations but I'm not sure what they are for. It could be that the work is not well motivated or just that it's not written to make the motivation and implications clear. Either way, I recommend rejection.

---

### Official Review · Reviewer_vGGV · 2021-11-07

**Correctness:** 2
**Technical Novelty And Significance:** 1
**Empirical Novelty And Significance:** 2
**Recommendation:** 3
**Confidence:** 3

**Main Review:**

I enjoyed a lot about this paper. It's very topical and I agree that studying representations is a more powerful and robust way to set up transfer learning than speculation/trial/empirical black-box analysis.

**Strengths**
1. I like the idea of taking metrics inspiration from neuroscience, and while many works in this space make only a tenuous connection between neuroscience and ML that doesn’t justify their claims, this one does a good job of not overstating its implications or forcing its hand. The metrics lend themselves intuitively and elegantly to the representation analysis outlined here
2. The experiments are thoughtful and interesting, and I think each piece of work in this paper is useful. Rather than doing several different experiments and simply listing them all, this paper justifies each one. Nice work, and (imo) not as common as it should be in our field!
3. The figures in this paper are incredibly informative. Particularly figures 6 and 7, but also the similarity matrices in 4 and 5: they're really informative, and 6 and 7 in particular are also easy to read and aesthetic.
4. For the most part, this paper is well-written. It's easy to follow while delivering lots of information. Compliments to the authors on writing a paper that lets you sit with the science instead of forcing you to puzzle over sentence structure.

**My critiques:**
This paper has great experiments and is one of the few where I liked the results section more than the prior sections (though that makes sense with an analytical paper). However, I don't think it makes or supports clear claims. This is both a content and a presentation issue, and it is the reason I do not recommend acceptance at this moment. I realize that this paper is meant to be exploratory and open new avenues, but even an introductory paper should come to one or a few concrete, robust points and that doesn't quite happen here. That said, I think this work could be turned into two or three (still introductory, exploratory) papers with some more work done on each of its preliminary findings and I highly encourage this direction. I also note that this may be a matter of my research taste and other reviewers may feel differently.

Specifics:
1. This paper doesn't make clear claims. Until section 3, I know what problem it is working on but I have no idea what answer it is going to come up with. Even if it is exploratory, I should know which findings the exploration yielded once I'm done with the introduction because otherwise I have no idea what to look for and have to piece together the unifying threads of analysis myself. Points where this shows up:
- The first sections only talk about exploration and problem, but the conclusion talks about several findings (e.g. unsupervised objectives can capture some but not all the representations of RL) that I learned about halfway through section 3 or later - more than halfway through the paper
- The claims made in the conclusion appear to be that RL is a general outlier, all objectives have representations that become less similar at later layers, unsupervised objectives can capture some but not all of the representations of RL, and the RL-trained visual encoder seems to drive long-term behavioral planning. However, so much of section 3.4 and 3.5 is about how RL representations are sparse that I would expect more discussion of it in Section 4 than "RL is an outlier". So I don't know what to expect, I form expectations based on the distribution of detail in section 3, and then those expectations are violated at the end.
- There are interesting details thrown at the reader, e.g. unsupervised tasks representing pixel space and supervised tasks reaching action space. But this doesn't get enough follow-up experimentation, and it's hard to tell if it's surprising given the nontrivial difference in performance. Another example is the fact that RLRod and UnsVar have R2 look more like pixel space than R1.
- The paper seems to focus on RL and its distinctiveness from supervised and unsupervised learning, but the title and introductory sections don't really suggest this. The problem setup talks about RL in particular, but we aren't told that the results will focus on it; only that it's a good motivator.

2. By the end, we have some claims (what I listed above). However, they seem somewhat speculative and I don't think there is enough evidence behind most of them. Examples:
- As I said, much of the attention is on RL, but there's only one RL objective and several supervised and unsupervised objectives. I would recommend more RL experimentation with alternative tasks, even if not as high-dimensional (in fact this might strengthen the promising results about RL representation dimensionality).
- The claim about RL having neurons that respond to one and only one image is major, especially since it's true in early layers and becomes less true with later layers. I think the evidence supports this hypothesis, but I am not sure if it's enough when all we know is that the representation is sparse and high dimensional. I may not be knowledgeable enough to know what's required to convince of this claim, but it would help to have more exposition and I think more evidence
- It's interesting that RLRod and UnsVar have R2 look more like pixel space than R1, but the analysis is "this odd effect seems to be the result of the maze task being quite distinct in pixel space, but this distinction doesn't emerge until later." That's a strong causal claim for a surprising result, and we don't have nearly enough evidence to suspect that because it's not clear mathematically how it would happen.
- The claim that proprioception is necessary for reliable action modeling is definitely suggested, but not quite demonstrated with only one experiment. The offshoot, that not only is proprioception important but visual information becomes useful longer-term, is *very* speculative. The explanation brings up objectives that haven't been tested. The discussion then reports this as one of ~four primary findings even though it's a one line suggestion in the analysis, which once again feels insufficient.

**Presentational notes:**
- Figure 4 and figure 5 are very useful and informative. That said, I find correlation matrices a little difficult to visually parse, so better signposting (callouts, highlights, more labelling) could be helpful
- I would appreciate visual examples if possible. For example, renders of the activations to show pixel vs. action representation could be helpful to give the reader an intuitive sense for the claims that these are surprising results and not just matters of less learning.

**Nit:**
Section 3.4 refers to figure 7 but I think it's meant to refer to figure 6. This was quite a confusing error (and I may be wrong).

This paper delivers analysis as though it is happening in real-time, rather than before writing. Even if that is what happened (as it does), the paper still shouldn't sound like that because it's confusing and forces the reader to do work that 1) they shouldn't have to do and 2) they will not be as good at as the authors. This paper has interesting, promising content and with more work could yield two or three interesting papers about the long-term use of visual information, the sparsity of of RL representations, or the comparative expressiveness of RL and unsupervised learning representations. These themselves would open avenues of investigation while still being complete papers; as it stands, the work currently feels more like research notes and less like a checkpoint, let alone finished work. I highly recommend expanding and submitting to a later venue, where I expect multiple works stemming from this to do well.

**Summary Of The Paper:**

This work identifies the important problem of transfer learning for problems that require deep learning due to high-dimensional inputs. It notes that transfer learning approaches often cannot transfer very far across learning objectives and are judged empirically as black boxes, making for brittle approaches. It proposes to study internal representations to assess potential for transfer.

This study trains several supervised and unsupervised, and one reinforcement, tasks with the same data and ResNet architecture. It uses representational similarity analysis (inter-task similarity similarity of intra-task dissimilarity), sparsity analysis (neuroscience-sourced measure of response similarity between neurons), and dimensionality analysis (PCA) to assess the representations; these metric choices are taken from neuroscience literature.

RSA shows that representations across different tasks diverge with layer depth (except in a few cases, though the paper doesn't call these out as clearly); the same is seen within different instantiations of the same task in all but three cases. RLRod differs the most. Sparseness analysis shows that RLRod has the most internal dissimilarity in responses to various inputs, and dimensionality analysis shows that it is also the highest dimensional. The paper speculates that this might be due to a one-image-per-neuron representation emerging even in early layers. To test whether the high dimensionality in RLRod is purely due to a higher-dimensional output (38-dim action), the paper trains encoders to predict action and finds that the visual inputs are pretty useless and proprioceptions are somewhat helpful, indicating the value of proprioceptive data.

The discussion lists some main claims: representations become less similar at later layers, unsupervised objectives can capture some but not all of the representations of RL, the RL-trained visual encoder seems to drive long-term behavioral planning; then suggests the potential for more study to determine promising transfers. It also postulates value for the neuroscience community, since sparse representations are prevalent in biological intelligence and the similarity may be indicative of similar objectives in artificial and biological intelligence.

**Summary Of The Review:**

Strengths:
1. Good, well-scoped inspiration from neuroscience
2. Thoughtful and interesting experiments
3. Very informative figures, several of which are also aesthetically appealing and clear
4. Overall very interesting problem and motivation, plus a well-written paper

My critiques:
1. Lack of clear claims, especially at the beginning, making it difficult to have a good lens with which to view the results - see examples
2. Claims appear by the end, but they are somewhat speculative and disparate - see examples

Presentation notes:
- The figures are very rich and useful. Better signposting in them would help the reader really get the value of these experiments
- Visual examples of activations for certain points would help build intuition for why these might be surprising results

---

### Author Response · Authors · 2021-11-16
**Response to reviewers**

We'd like to thank the reviewers for their close reading of our work and the helpful feedback they provided. We will take their comments into account as we go forward with this research direction, but will not be updating our ICLR submission.

---

### Decision · Program_Chairs · 2022-01-20

**Decision:**

Reject

**Comment:**

This paper explores the representations that are learned by the same network, on the same data, but with different objectives/tasks (RL, supervised, unsupervised). Though the reviewers were positive about some aspects of the paper, the reviews were generally low (3,3,3,6) and indicated rejection. The principle recurring theme in the reviews as to why this was a rejection was the lack of clear motivations/implications. The authors decided not to submit an updated version of the paper. As such, this was a reject decision.